# Sparse Quadratic Optimisation over the Stiefel Manifold with Application to Permutation Synchronisation

**Florian Bernard**
TU Munich, University of Bonn

**Daniel Cremers**
TU Munich

**Johan Thunberg**
Halmstad University

## Abstract

We address the non-convex optimisation problem of finding a sparse matrix on the Stiefel manifold (matrices with mutually orthogonal columns of unit length) that maximises (or minimises) a quadratic objective function. Optimisation problems on the Stiefel manifold occur for example in spectral relaxations of various combinatorial problems, such as graph matching, clustering, or permutation synchronisation. Although sparsity is a desirable property in such settings, it is mostly neglected in spectral formulations since existing solvers, e.g. based on eigenvalue decomposition, are unable to account for sparsity while at the same time maintaining global optimality guarantees. We fill this gap and propose a simple yet effective sparsity-promoting modification of the Orthogonal Iteration algorithm for finding the dominant eigenspace of a matrix. By doing so, we can guarantee that our method finds a Stiefel matrix that is globally optimal with respect to the quadratic objective function, while in addition being sparse. As a motivating application we consider the task of permutation synchronisation, which can be understood as a constrained clustering problem that has particular relevance for matching multiple images or 3D shapes in computer vision, computer graphics, and beyond. We demonstrate that the proposed approach outperforms previous methods in this domain.

## 1 Introduction

We are interested in optimisation problems of the form

$$\underset{U \in \mathrm{St}(m,d)}{\arg\max} \; f(U) \; \text{ with } \; f(U) = \mathrm{tr}(U^T W U) \; \text{ and } \; \mathrm{St}(m,d) := \{X \in \mathbb{R}^{m \times d} : X^T X = \mathbf{I}_d\}, \quad (1)$$

where $\mathbf{I}_d$ is the identity matrix of dimension $d$, $W \in \mathbb{R}^{m \times m}$ and the set $\mathrm{St}(m,d)$ denotes the Stiefel manifold ($m \geq d$). Throughout the paper, w.l.o.g. we consider a maximisation formulation and we assume that $W$ is a symmetric and positive semidefinite matrix (see Lemma 1 in Sec. 2). Despite the non-convexity of Problem (1), problems of this form can be solved to global optimality based on the eigenvalue decomposition, i.e. by setting $U^* = V_d$, where $V_d \in \mathbb{R}^{m \times d}$ denotes an orthogonal basis of eigenvectors corresponding to the $d$ largest eigenvalues of $W$, see Lemma 2. The fact that we can efficiently find global optima of Problem (1) makes it a popular relaxation formulation of various difficult combinatorial problems. This includes spectral relaxations [43, 26, 17] of the NP-hard quadratic assignment problem (QAP) [35], spectral clustering [33, 44], or spectral permutation synchronisation [34, 39, 31]. Yet, a major difficulty of such approaches is to discretise the (relaxed) continuous solution in order to obtain a feasible solution of the original (discrete) problem. For example, in the QAP one optimises over the set of permutation matrices, in clustering one optimises over the set of matrices with rows comprising canonical basis vectors, and in permutation synchronisation one optimises over multiple permutation matrices that are stacked into a large block

matrix. Often such combinatorial sets can be characterised by Stiefel matrices that are *sparse* – for example, the set of (signed) $d \times d$ permutation matrices can be characterised by matrices in $\mathrm{St}(d, d)$ that have exactly $d$ non-zero elements, whereas any other element in $\mathrm{St}(d, d)$ that has more than $d$ non-zero elements is not a signed permutation matrix.

The most common approach to obtain sparse solutions in an optimisation problem is to integrate an explicit sparsity-promoting regulariser. However, a major hurdle when considering quadratic optimisation problems over the Stiefel manifold is that the incorporation of sparsity-promoting terms is not compatible with solutions based on eigenvalue decomposition, so that desirable global optimality guarantees are generally no longer maintained.

Instead, we depart from the common path of integrating explicit regularisers and instead exploit the orthogonal-invariance present in Problem (1) to assure sparsity. To be more specific, for any orthogonal matrix $Q \in \mathbb{O}(d) := \mathrm{St}(d, d)$ it holds that $f(U) = \mathrm{tr}(U^T W U) = \mathrm{tr}(U^T W U Q Q^T) = \mathrm{tr}((UQ)^T W (UQ)) = f(UQ)$. Hence, if $\bar{U} \in \mathrm{St}(m, d)$ is a solution to Problem (1), so is $\bar{U}Q \in \mathrm{St}(m, d)$ for any $Q \in \mathbb{O}(d)$. The subspace $\mathrm{im}(\bar{U}Q)$ that is spanned by $\bar{U}Q$ for a given $\bar{U} \in \mathrm{St}(m, d)$ (and arbitrary $Q \in \mathbb{O}(d)$) is equal to the subspace $\mathrm{im}(\bar{U})$. Motivated by this observation we utilise orthogonal-invariance in order to find a solution $U^* \in \{U \in \mathrm{St}(m, d) : \mathrm{im}(U) = \mathrm{im}(\bar{U})\}$ such that $U^*$ is sparse. To this end, we build upon the additional degrees of freedom due to $Q \in \mathbb{O}(d)$, which allows to rotate a given solution $\bar{U} \in \mathrm{St}(m, d)$ to a sparser representation $U^* = \bar{U}Q \in \mathrm{St}(m, d)$ – most notably, while remaining a globally optimal solution to Problem (1).

**Main contributions.** We summarise our main contributions as follows: (i) For the first time we propose an algorithm that exploits the orthogonal-invariance in quadratic optimisation problems over the Stiefel manifold while simultaneously accounting for sparsity in the solution. (ii) Despite its simplicity, our algorithm is effective as it builds on a modification of the well-established Orthogonal Iteration algorithm for finding the most dominant eigenspace of a given matrix. (iii) Our algorithm is guaranteed to converge to the dominant subspace with the same convergence rate as the original Orthogonal Iteration algorithm, and our solution constitutes a global optimiser of Problem (1). (iv) We experimentally confirm the efficacy of our approach in the context of the permutation synchronisation problem.

## 2    Preliminaries & Related Work

In this section we clarify our assumptions, introduce additional preliminaries, and provide reference to related work. Let $\lambda_1, \lambda_2, \ldots, \lambda_m$ be the eigenvalues of $W \in \mathbb{R}^{m \times m}$ ordered decreasingly. We impose the following assumption on $W$:

**Assumption 1** (Separated eigenspace)**.** *We assume that $\lambda_d > \lambda_{d+1}$.*

Throughout the paper we also assume that $W$ is symmetric and positive semidefinite (p.s.d.), which, however, is not a restriction as the following straightforward result indicates:

**Lemma 1** (Symmetry and positive semidefiniteness)**.** *In Problem* (1)*, if $W$ is not symmetric and not p.s.d. there is an equivalent optimisation problem (i.e. with the same optimisers) where $W$ has been replaced by a symmetric p.s.d. matrix $\tilde{W}$.*

*Proof.* See Appendix. □

Next, we define the notion of a dominant subspace and convergence of matrix sequences to such a subspace.

**Definition 1** (Dominant invariant subspace)**.** *The $d$-dimensional dominant invariant subspace (or dominant subspace in short) of the matrix $W \in \mathbb{R}^{m \times m}$ is defined as the subspace $\mathrm{im}(V_d) \subseteq \mathbb{R}^m$, where $V_d \in \mathrm{St}(m, d)$ is the matrix whose columns are formed by the $d$ eigenvectors corresponding to the $d$ largest eigenvalues of $W$.*

**Definition 2** (Convergence)**.** *We say that a sequence of matrices $\{U_t\}$ converges to the dominant subspace of $W$ if $\lim_{t \to \infty} \|V_d V_d^T U_t - U_t\| = 0$.*

There exists a close relation between the dominant subspace of $W$ and solutions of Problem (1):

**Lemma 2** (Solution to Problem (1)). *Problem* (1) *is solved for any matrix $U^*$ that forms an orthogonal basis for the $d$-dimensional dominant subspace of $W$, i.e. $U^* \in \{U \in \mathrm{St}(m, d) : \mathrm{im}(U) = \mathrm{im}(V_d)\}$.*

*Proof.* See Appendix. $\square$

**Basic algorithms for computing eigenvectors.** In order to find the dominant subspace of $W$ we can use algorithms for finding eigenvectors. The Power method [21] is an efficient way for finding the single most dominant eigenvector of a given matrix, i.e. it considers the case $d = 1$. It proceeds by iteratively updating a given initial $v_0 \in \mathrm{St}(m, 1)$ based on the update $v_{t+1} \leftarrow W v_t / \|W v_t\|$. In order to find the $d$ most dominant eigenvectors, one can consider the Orthogonal Iteration algorithm [21], which generalises the Power method to the case $d > 1$. The algorithm proceeds by repeatedly computing $V_{t+1} R_{t+1} \leftarrow W V_t$ based on the (thin) QR-decomposition of $W V_t$, where $V_t \in \mathrm{St}(m, d)$ and $R_t \in \mathbb{R}^{d \times d}$ is upper triangular. For $t \to \infty$ the sequence $\{V_t\}$ converges (under mild conditions on the intial $V_0$) to the dominant invariant subspace as long as $|\lambda_d| > |\lambda_{d+1}|$, see Thm. 8.2.2 in [21].

**Sparse Stiefel optimisation.** There are numerous approaches for addressing general optimisation problems over the Stiefel manifold, including generic manifold optimisation techniques (e.g. [3, 14]), or Stiefel-specific approaches (e.g. [30, 46]). In the following we will focus on works that consider *sparse* optimisation over the Stiefel manifold that are most relevant to our approach. In order to promote sparse solutions, sparsity-inducing regularisers can be utilised, for example via the minimisation of the (non-convex) $\ell_p$-'norm' for $0 \leq p < 1$, or the (convex) $\ell_1$-norm [36]. However, the non-smoothness of such regularisers often constitutes a computational obstacle.

The optimisation of non-smooth functions over the Stiefel manifold has been considered in [16], where the sum of a non-convex smooth and a convex non-smooth function is optimised via a proximal gradient method. In [27], the authors consider the optimisation of a sum of non-smooth weakly convex functions over the Stiefel manifold using Riemannian subgradient-type methods. Yet, in practice often differentiable surrogates of non-smooth sparsity-promoting terms are considered [40, 29, 38, 15]. Instead of minimising $\ell_p$-'norms' with $0 \leq p \leq 1$, on the Stiefel manifold one may instead choose the *maximisation* of $\ell_p$-norms with $p > 2$, such as the $\ell_3$-norm [47], or the $\ell_4$-norm [49]. Further motivation for sparsity promoting higher-order norms in this context can be found in [37, 50, 28].

Optimisation over the Stiefel manifold has a close connection to optimisation over the Grassmannian manifold [3]. There are numerous approaches for Grassmannian manifold optimisation (e.g. [19, 18]), including sparse optimisation via an $\ell_1$-norm regulariser [45] and the optimisation of non-convex and non-smooth objective functions via a projected Riemannian subgradient method [52]. In our case, due to the rotation-invariance of the objective $f(U) = f(UQ)$ for any $Q \in \mathbb{O}(d)$ in Problem (1), finding the dominant subspace of $W$ could also be posed as an optimisation problem over the Grassmannian manifold. However, we are not only interested in identifying this subspace (which can for example be done via the Orthogonal Iterations algorithm [21], or via Grassmann-Rayleigh Quotient Iterations [4]), but we want to find a specific choice of coordinates for which the representation of the subspace is sparse.

## 3 Proposed Method for Sparse Quadratic Optimisation over the Stiefel

**Input:** $W \in \mathbb{R}^{m \times m}$, $U_0 \in \mathbb{R}^{m \times d}$
**Output:** $U^*$
**Initialise:** $t \leftarrow 0$
**repeat**
$\quad \mid \quad U_{t+1} R_{t+1} \leftarrow W U_t Z(U_t)$ // `unique QR-decomposition`
**until** *convergence*
$U^* \leftarrow U_{t+1}$
**Algorithm 1:** Overview of our proposed algorithm. In terms of convergence properties our algorithm is equivalent to the Orthogonal Iteration algorithm and thus produces a $U^*$ in the dominant subspace of $W$. However, our modification introduces the matrix $Z(U_t)$ with the purpose of promoting sparsity of the solution (see Sec. 3.2 how we choose $Z(U_t)$).

In this section we introduce our Algorithm 1 that can be seen as a modification of the popular Orthogonal Iteration algorithm. The main difference is that we iteratively weigh the matrix $U_t \in \mathrm{St}(m,d)$ by a matrix $Z(U_t) \in \mathbb{R}^{d \times d}$, where the matrix function $Z$ maps onto the set of full rank matrices. The purpose of the matrix $Z(U_t)$ (see Sec. 3.2 for our specific choice) is to promote sparsity. Intuitively, we characterise sparsity as having few elements that are large, whereas most elements are close to zero, which is formally expressed in (4).

First, we focus on the overall interpretation of our approach: We want to ensure that we retrieve a $U^*$ that is a global maximiser of Problem (1), which, in some relaxed sense, is also sparse. However, we cannot in general augment the objective function in Problem (1) with an explicit sparsity regulariser and expect that the respective solution is still a maximiser of the original objective. Instead we steer the solution towards being more sparse by weighing our matrix of interest $U_t$ with the matrix $Z(U_t)$.

## 3.1 Convergence

We start by ensuring that, under the assumption that $Z_t(U_t)$ is full rank, the sequence $\{U_t\}$ generated by Algorithm 1 converges to the dominant subspace of $W$. This can straightforwardly be shown by using the following result.

**Lemma 3.** *Consider the two algorithms:*

> **Algorithm A:** $V_{t+1}R_{t+1} \leftarrow WV_tX_t$,
>
> **Algorithm B:** $\bar{V}_{t+1}\bar{R}_{t+1} \leftarrow W\bar{V}_t$,

*where $X_t \in \mathbb{R}^{d \times d}$ is full rank, and for each of the two left-hand sides above the two matrices in the product are obtained by the unique (thin) QR-factorisation of the corresponding right-hand side, where $R_{t+1}$ and $\bar{R}_{t+1}$ are upper triangular with positive diagonal. Now, if $V_t = \bar{V}_t\bar{Q}_t \in \mathrm{St}(m,d)$, where $\bar{Q}_t \in \mathbb{O}(d)$, then $V_{t'}$ is equal to $\bar{V}_{t'}$ up to rotation from the right for all $t' > t$.*

*Proof.* It holds that $V_{t+1}R_{t+1} = WV_tX_t = W\bar{V}_t\bar{Q}_tX_t = \bar{V}_{t+1}(\bar{R}_{t+1}\bar{Q}_tX_t) = \bar{V}_{t+1}\tilde{Q}_{t+1}\tilde{R}_{t+1}$, where $\tilde{Q}_{t+1}$ and $\tilde{R}_{t+1}$ are the two matrices in the QR-factorisation of the matrix $(\bar{R}_{t+1}\bar{Q}_tX_t)$. This means that $V_{t+1} = \bar{V}_{t+1}\tilde{Q}_{t+1}$. This is due to the uniqueness of the QR-factorisation, so that also $R_{t+1} = \tilde{R}_{t+1}$. Now the results follows readily by using induction. $\square$

Thus, if $Z_t(U_t)$ is full rank for each $t$, we can identify our Algorithm 1 with Algorithm A in Lemma 3, and conclude that the sequence $\{U_t\}$ converges to the dominant subspace of $W$ as $t \to \infty$. This is because Algorithm B in Lemma 3 corresponds to the Orthogonal Iteration algorithm, which is known to converge to the dominant subspace of $W$ [21].

We continue by investigating the behaviour at the limit where the columns of $U_t$ are in the dominant subspace of $W$. Let us consider the update $U_{t+1}R_{t+1} \leftarrow WU_tZ(U_t)$ of Algorithm 1, and further assume that the $d$ largest eigenvalues of $W$ are equal, which is for example the case for synchronisation problems assuming cycle consistency (cf. Sec. 4).

**Lemma 4.** *Assume that the $d$ largest eigenvalues of the p.s.d. matrix $W$ are all equal to one[1] and strictly larger than the other eigenvalues, and assume that the columns of $U_t \in \mathrm{St}(m,d)$ are contained in the dominant subspace of $W$. Provided $Z(U_t)$ is full rank, it holds that*

$$WU_tZ(U_t) = U_tZ(U_t) = U_{t+1}R_{t+1} = U_tQ_{t+1}R_{t+1},$$

*where $U_{t+1}R_{t+1}$ is the unique QR-factorisation of $U_tZ(U_t)$ and $Q_{t+1}R_{t+1}$ is the unique QR-factorisation of $Z(U_t)$.*

*Proof.* Since all the $d$ largest eigenvalues of $W$ are equal to one and the columns of $U_t$ are in the dominant subspace it holds that $WU_t = U_tU_t^TU_t = U_t$, which explains the first equality. While the second equality follows by definition, the third equality remains to be proven. Let $Q_{t+1}\tilde{R}_{t+1}$ be the unique QR-decomposition of $Z(U_t)$. We want to show that $U_{t+1} = U_tQ_{t+1}$ and $R_{t+1} = \tilde{R}_{t+1}$. From the QR-decomposition of $Z(U_t)$ it follows that $U_tZ(U_t) = U_tQ_{t+1}\tilde{R}_{t+1}$, while from the

---

[1]If they are all equal to a value $\lambda_1 \neq 1$, we can w.l.o.g. consider $\lambda_1^{-1}W$ in place of $W$, since the optimiser of Problem (1) is invariant to scaling $W$.

QR-decomposition of $U_t Z(U_t)$ it follows that $U_t Z(U_t) = U_{t+1} R_{t+1}$. The uniqueness of the QR-decomposition implies $U_t Q_{t+1} = U_{t+1}$ and $\tilde{R}_{t+1} = R_{t+1}$. $\qquad\square$

Hence, under the assumptions in Lemma 4, the columns of $U_{t'}$ span the dominant subspace of $W$ for any $t' \geq t$. The update in Algorithm 1 reduces to the form

$$Q_{t+1} R_{t+1} \leftarrow Z(U_t), \tag{2}$$
$$U_{t+1} \leftarrow U_t Q_{t+1}, \tag{3}$$

i.e. updating $U_t$ to obtain $U_{t+1}$ simplifies to multiplication of $U_t$ with the $Q$-matrix from the QR-factorisation of $Z(U_t)$.

## 3.2    Choosing $Z(U_t)$ to Promote Sparsity

We now turn our attention to the choice of $Z(U_t)$ in Algorithm 1. We know that, as long as $Z(U_t)$ is full rank in each iteration $t$, the columns of $U_t$ converge to the dominant subspace of $W$. In addition to our objective function $f$ in Problem (1), we now introduce the secondary objective

$$g(U) = \sum_{i=1}^{m} \sum_{j=1}^{d} (U_{ij})^p = \text{tr}(U^T(U.^{\wedge(p-1)})), \tag{4}$$

where for $p$ being a positive integer the notation $U.^{\wedge p}$ means to raise each element in $U$ to the power $p$. For $U \in \text{St}(m, d)$ and $p$ larger than 2, the maximisation of $g$ promotes *sparsity in a relaxed sense*. To be specific, if $p$ is odd, a few larger elements (with value close to 1) lead to a larger value of $g$ compared to many smaller elements (with value close to 0), so that sparsity and non-negativity are simultaneously promoted. Analogously, if $p$ is even, a few elements with value closer to $\pm 1$ lead to a larger value of $g$ compared to many smaller elements close to 0.

Let us consider the maximisation of

$$g(UQ) = \text{tr}(h(U, Q)) \tag{5}$$

with respect to $Q$, where $h(U, Q) = (UQ)^T((UQ).^{\wedge(p-1)})$, $Q \in \mathbb{O}^d$, and $U \in \text{St}(m, d)$. This means that we want to rotate $U$ by $Q$ in such a way that the secondary objective $g$ is maximised after the rotation. By applying the rotation from the right, we ensure that the columns $U$ span the same space after the rotation, but a higher objective value is achieved for the secondary objective $g$ in (4).

The extrinsic gradient with respect to $Q$ (for $Q$ relaxed to be in $\mathbb{R}^{d \times d}$) of $g(UQ)$ at $Q = \mathbf{I}$ is $p \cdot h(U, \mathbf{I})$. Thus the (manifold) gradient of $g(UQ)$ at $Q = \mathbf{I}$ is given by the projection of the extrinsic gradient at $Q = \mathbf{I}$ onto the tangent space, which reads

$$\nabla_Q g(UQ)|_{Q=\mathbf{I}} = p(h(U, \mathbf{I}) - h^T(U, \mathbf{I})). \tag{6}$$

For a small step size $\alpha$, we can perform a first-order approximation of gradient ascent if we choose $Z = \mathbf{I} + \alpha p(h(U, \mathbf{I}) - h^T(U, \mathbf{I}))$, and then update $U$ in terms of the Q-matrix of the QR-factorisation of $Z$. Now, let us assume that we investigate the behaviour at the limit under the assumptions in Lemma 4, i.e. that the $d$ largest eigenvalues of $W$ are equal. In this context the Q-matrix of the QR-factorisation is a retraction [3] and serves as a first-order approximation of the exponential map. With that, the updates become

$$Q_{t+1} R_{t+1} \leftarrow \mathbf{I} + \alpha_t (h(U_t, \mathbf{I}) - h^T(U_t, \mathbf{I})), \tag{7}$$
$$U_{t+1} \leftarrow U_t Q_{t+1}. \tag{8}$$

This choice of the matrix $Z(U_t)$ ensures that it is full rank, since adding the identity matrix and a skew-symmetric matrix results in a full rank matrix. Hence, for sufficiently small step size $\alpha$, we iteratively move in an ascent direction on $\mathbb{O}^d$ by utilising the QR-retraction.

## 4    Application to Permutation Synchronisation

### 4.1    Permutation Synchronisation in a Nutshell

Permutation synchronisation is a procedure to improve matchings between multiple objects [23, 34], and related concepts have been utilised to address diverse tasks, such as multi-alignment [7, 5,

25], multi-shape matching [22, 24, 20], multi-image matching [51, 42, 9, 13, 12], or multi-graph matching [48, 6, 41], among many others. Permutation synchronisation refers to the process of establishing cycle consistency in the set of pairwise permutation matrices that encode correspondences between points in multiple objects. In computer vision there is a typical application where the points are feature descriptors or key-points, and the objects are images, as shown in Fig. 2.

Let $k$ denote the number of objects, where each object $i$ contains $m_i$ points. For $\mathbf{1}_p$ being a $p$-dimensional vector of all ones, and vector inequalities being understood in an element-wise sense, let $P_{ij} \in \mathbb{P}_{m_i m_j} := \{X \in \{0,1\}^{m_i \times m_j} : X\mathbf{1}_{m_j} \leq \mathbf{1}_{m_i}, \mathbf{1}_{m_i}^T X \leq \mathbf{1}_{m_j}^T\}$ be the partial permutation matrix that represents the correspondence between the $m_i$ points in object $i$ and the $m_j$ points in object $j$. In the case of *bijective* permutations, the set of pairwise permutations $\mathcal{P} := \{P_{ij}\}_{i,j=1}^k$ is said to be cycle-consistent if for all $i, j, \ell$ it holds that $P_{i\ell} P_{\ell j} = P_{ij}$.

We define the set of partial permutation matrices with full row-rank as $\overline{\mathbb{P}}_{m_i d} := \{X \in \mathbb{P}_{m_i d} : X\mathbf{1}_d = \mathbf{1}_{m_i}\}$, where $d$ denotes the total number of distinct points across all objects. Cycle consistency is known to be equivalent to the existence of so-called object-to-universe matchings $\mathcal{U} := \{P_i \in \overline{\mathbb{P}}_{m_i d}\}_{i=1}^k$ such that for all $i, j$ we can write $P_{ij} = P_i P_j^T$ (see [23, 34] for details). The object-to-universe characterisation of cycle consistency is also valid for the case of non-bijective (i.e. partial) permutations (see [42, 8] for details).

Given the noisy (i.e. not cycle-consistent) set of pairwise permutations $\mathcal{P} = \{P_{ij}\}_{i,j=1}^k$, permutation synchronisation can be phrased as the optimisation problem

$$\underset{\{P_i \in \overline{\mathbb{P}}_{m_i d}\}}{\arg\max} \sum_{i,j} \operatorname{tr}(P_{ij}^T P_i P_j^T) \quad \Leftrightarrow \quad \underset{P \in \mathbb{U}}{\arg\max} \ \operatorname{tr}(P^T W P), \qquad (9)$$

where for $m := \sum_i m_i$ we define the set $\mathbb{U} := \overline{\mathbb{P}}_{m_1 d} \times \ldots \times \overline{\mathbb{P}}_{m_k d} \subset \mathbb{R}^{m \times d}$, the $(m \times d)$-dimensional block matrix $P = [P_1^T, \ldots, P_k^T]^T$, and the block matrix $W := [P_{ij}]_{ij} \in \mathbb{R}^{m \times m}$ in order to allow for a compact matrix representation of the objective.

## 4.2 Proposed Permutation Synchronisation Approach

The core idea of existing spectral permutation synchronisation approaches [34, 39, 31] is to replace the feasible set $\mathbb{U}$ in Problem (9) with the Stiefel manifold $\operatorname{St}(m, d)$, so that we obtain an instance of Problem (1). We utilise Algorithm 1 to obtain a (globally optimal) solution $U^* \in \operatorname{St}(m, d)$ of this spectral formulation. In our algorithm we choose $p = 3$ to promote sparsity and non-negativity in the resulting $U^*$ via the function $g(U) = \operatorname{tr}(U^T (U.^{\wedge 2})) = \sum_{i=1}^m \sum_{j=1}^d U_{ij}^3$. With that, in addition to $U^*$ being an orthogonal matrix, it contains few large elements that are close to 1 and many smaller elements that are close to 0. As such, we can readily project the matrix $U^*$ onto the set $\mathbb{U}$ in terms of a Euclidean projection. The Euclidean projection is given by projecting each of the $k$ blocks of $U^* = [U_1^{*T}, \ldots, U_k^{*T}]^T$ individually onto the set of partial permutations, i.e. $P_i = \operatorname{proj}_{\overline{\mathbb{P}}_{m_i d}}(U_i^*) = \arg\max_{X \in \overline{\mathbb{P}}_{m_i d}} \operatorname{tr}(X^T U_i^*)$, see e.g. [8], which amounts to a (partial) linear assignment problem [32, 11] that we solve based on the efficient implementation in [10].

## 4.3 Experimental Results

We experimentally compare our proposed approach with various methods for permutation synchronisation and perform an evaluation on both real and synthetic datasets. In particular, our comparison includes two existing spectral approaches, namely MATCHEIG [31] and SPECTRAL [34], where for the latter we use the efficient implementation from the authors of [51]. In addition, we also compare against the alternating minimisation method MATCHALS [51], and against the non-negative matrix factorisation approach NMFSYNC [8]. To emphasise that the methods MATCHEIG and MATCHALS do not guarantee cycle consistency (but instead aim to improve the initial matchings), in all plots we show results of respective methods as dashed lines. We use the fscore to measure the quality of obtained multi-matchings, which is defined as the fraction $f = \frac{2 \cdot p \cdot r}{p+r}$, where $p$ and $r$ denote the precision and recall, respectively. All experiments are run on a Macbook Pro (2.8 GHz quad core i7, 16 GB RAM), where for $\epsilon = 10^{-5}$ we use $f(U_t)/f(U_{t+1}) \geq 1 - \epsilon$ as convergence criterion in Algorithm 1, and a step size of $\alpha_t = \|h(U_t, \mathbf{I}) - h^T(U_t, \mathbf{I})\|_\infty^{-1}$ in (7).

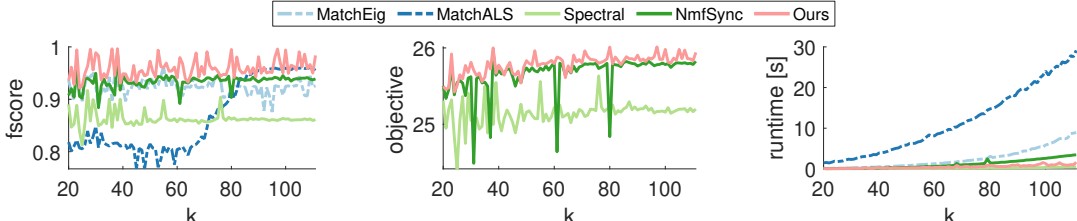

Figure 1: Quantitative results on the CMU house sequence in terms of the fscore (↑), objective value of Problem (9) (↑), and runtime (↓). The individual instances of permutation synchronisation problems vary along the horizontal axis. Methods that do not guarantee cycle consistency are shown as dashed lines.

**Real data.** In this experiment we use the CMU house image sequence [1] comprising 111 frames within the experimental protocol of [34]. We generate a sequence of permutation synchronisation problem instances with a gradually increasing number of objects $k$ by extracting respective pairwise matchings for $k$ objects. To this end, we vary $k$ from 20 to 111 and sample the pairwise matchings evenly-spaced from the $111 \times 111$ pairwise matchings. Quantitative results in terms of the fscore, the objective value of Problem (9), and the runtimes are shown in Fig. 1, in which the individual problem instances vary along the horizontal axis. We can see that our proposed method dominates other approaches in terms of the fscore and the objective value (the reported objective values are divided by $k^2$ to normalise the scale in problems of different sizes), while being among the fastest. Note that we do not report the objective value for MATCHEIG and MATCHALS, since they do not lead to cycle-consistent matchings so that the obtained solution does not lie in the feasible set of Problem (9). Qualitative results of the matching between one pair of images for $k = 111$ are shown in Fig. 2. As expected, our proposed approach clearly outperforms the SPECTRAL baseline, since our method is guaranteed to converge to the same same subspace that is spanned by the spectral solution, while at the same time providing a sparser and less negative solution that is thereby closer to the feasible set $\mathbb{U}$.

**Synthetic data.** We reproduce the procedure described in [8] for generating synthetic instances for the synchronisation of partial permutations. Four different parameters are considered for generating a problem instance: the universe size $d$, the number of objects $k$ that are to be matched, the observation rate $\rho$, and the error rate $\sigma$ (see [8] for details). One of these parameters varies in each experimental setting, while the others are kept fixed. Each individual experiment is repeated 5 times with different random seeds. In Fig. 3 we compare the performance of MATCHEIG [31], MATCHALS [51], SPECTRAL [34], NMFSYNC [8] and OURS. The first row shows the fscore, and the second row the respective runtimes. Note that we did not run MATCHALS on the larger instances since the runtime is prohibitively long. The methods MATCHEIG and MATCHALS do not guarantee cycle consistency, and are thus shown as dashed lines. It can be seen that in most settings our method obtains superior performance in terms of the fscore, while being almost as fast as the most efficient methods that are also based on a spectral relaxation (MATCHEIG and SPECTRAL).

## 5 Discussion and Limitations

Our algorithm has several favourable properties: it finds a globally optimal solution to Problem (1), it is computationally efficient, it has the same convergence behaviour as the Orthogonal Iteration algorithm [21], and promotes (approximately) sparse solutions. Naturally, since our solution is globally optimal (with respect to Problem (1)) and thus converges to $\text{im}(V_d)$, the amount of sparsity that we can achieve is limited by the sparsest orthogonal basis that spans $\text{im}(V_d)$. In turn, we rather interpret sparsity in some looser sense, meaning that there are few elements that are large, whereas most elements are close to zero. Furthermore, since the secondary sparsity-promoting problem (with objective in (4)) is generally non-convex, we cannot guarantee that we attain its global optimum.

Since permutation synchronisation was our primary motivation it is the main focus of this paper. In the case of permutations, the assumption that the eigenvalues corresponding to the $V_d$ are equal (as we explain in Sec. 3.2) is reasonable, since this must hold for cycle-consistent bijective matchings. However, broadening the scope of our approach to other problems in which this assumption may

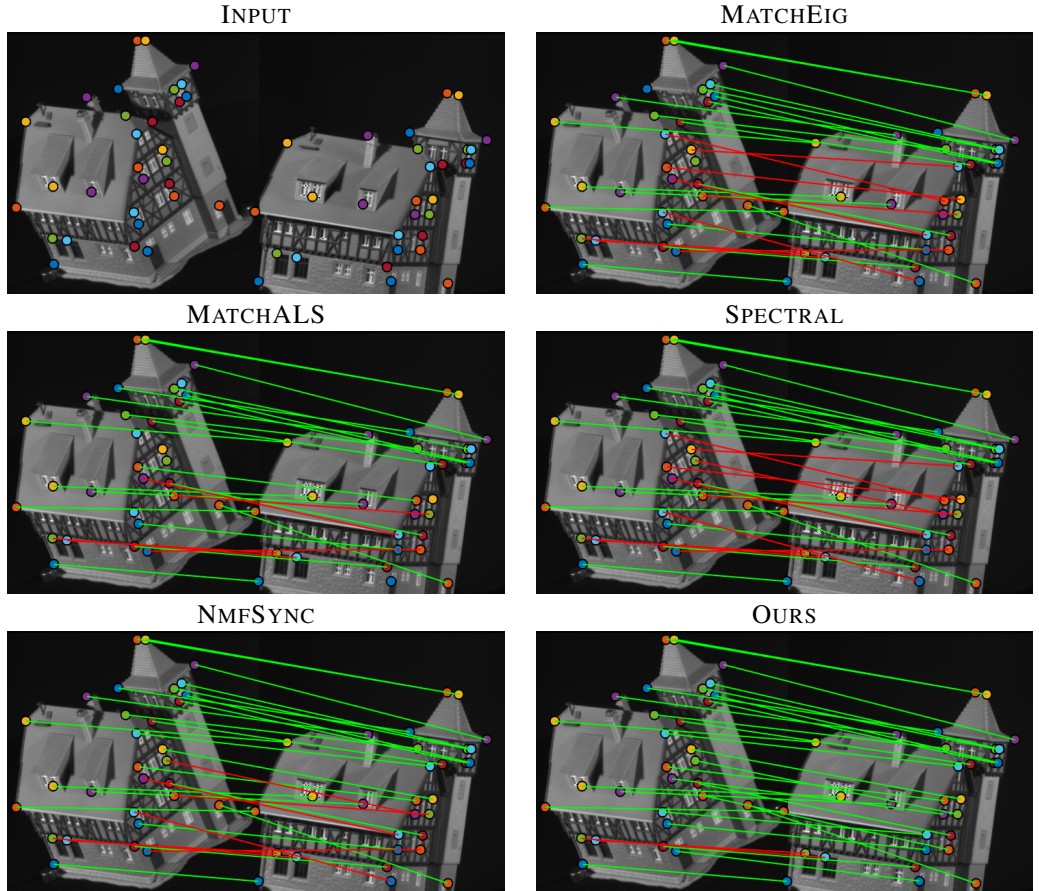

Figure 2: Comparison of matchings between the first and last image of the CMU house sequence obtained by several methods. The colour of the dots indicates the ground truth correspondence, and the lines show the obtained matchings (green: correct, red: wrong). Overall, our approach obtains the best matchings, see also Fig. 1.

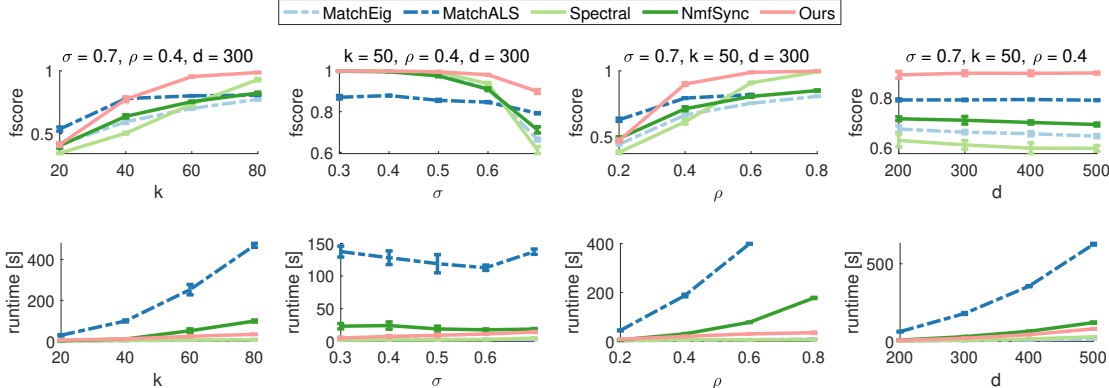

Figure 3: Experimental comparison of permutation synchronisation methods on synthetic data. Each column shows a different varying parameter. The first row shows the fscore ($\uparrow$) and the second row the runtime ($\downarrow$). Methods that do not guarantee cycle consistency are shown with dashed lines. Our method guarantees cycle consistency and leads to higher fscores in most settings.

not be valid will require further theoretical analysis and possibly a different strategy for choosing the matrix $Z(U_t)$. Studying the universality of the proposed method, as well as analysing different

convergence criteria and different matrix functions $Z(U_t)$ are open problems that we leave for future work.

The proposed method comprises a variation of a well-known iterative procedure for computing the dominant subspace of a matrix. The key component in this procedure is the QR-factorisation, which is differentiable almost everywhere and is therefore well-suited to be applied within a differentiable programming context (e.g. for end-to-end training of neural networks).

## 6   Conclusion

We propose an efficient algorithm to find a matrix that forms an orthogonal basis of the dominant subspace of a given matrix $W$ while additionally promoting sparsity and non-negativity. Our procedure inherits favourable properties from the Orthogonal Iteration algorithm, namely it is simple to implement, converges almost everywhere, and is computationally efficient. Moreover, our method is designed to generate sparser solutions compared to the Orthogonal Iteration algorithm. This is achieved by rotating our matrix of interest in each iteration by a matrix corresponding to the gradient of a secondary objective that promotes sparsity (and optionally non-negativity).

The considered problem setting is relevant for various spectral formulations of difficult combinatorial problems, such as they occur in applications like multi-matching, permutation synchronisation or assignment problems. Here, the combination of orthogonality, sparsity and non-negativity is desirable, since these are properties that characterise binary matrices such as permutation matrices. Experimentally we show that our proposed method outperforms existing methods in the context of partial permutation synchronisation, while at the same time having favourable a runtime.

**Broader Impact**

The key contribution of this paper is an effective and efficient optimisation algorithm for addressing sparse optimisation problems over the Stiefel manifold. Given the fundamental nature of our contribution, we do not see any direct ethical concerns or negative societal impacts related to our work.

Overall, there are numerous opportunities to use the proposed method for various types of multi-matching problems over networks and graphs. The problem addressed is highly relevant within the fields of machine learning (e.g. for data canonicalisation to faciliate efficient learning with non-Euclidean data), computer vision (e.g. for 3D reconstruction in structure from motion, or image alignment), computer graphics (e.g. for bringing 3D shapes into correspondence) and other related areas.

The true power of permutation synchronisation methods appears when the number of objects to synchronise is large. With large quantities of data becoming increasingly available, the introduction of our scalable optimisation procedure that can account for orthogonality – while promoting sparsity and non-negativity – is an important contribution to the synchronisation community on the one hand. Moreover, on the other hand, it has the potential to have an impact on more general non-convex optimisation problems, particularly in the context of spectral relaxations of difficult and large combinatorial problems.

**Acknowledgement**

JT was supported by the Swedish Research Council (2019-04769). We thank Marvin Eisenberger for helpful feedback.

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
