# Sparse Quadratic Optimisation over the Stiefel Manifold with Application to Permutation Synchronisation

**Florian Bernard**
TU Munich, University of Bonn

**Daniel Cremers**
TU Munich

**Johan Thunberg**
Halmstad University

## A  Proofs

### Proof of Lemma 1

*Proof.* If the matrix $W$ is not symmetric, we can split $W$ into the sum of a symmetric part $W_s$ and skew-symmetric part $W_a$. It holds that $\operatorname{tr}(U^T W_a U) = \frac{1}{2}\operatorname{tr}(U^T W_a U) + \frac{1}{2}\operatorname{tr}(U^T W_a^T U) = \frac{1}{2}\operatorname{tr}(U^T W_a U) - \frac{1}{2}\operatorname{tr}(U^T W_a U) = 0$. Further, if $W_s$ is not positive semidefinite, we can shift its eigenvalues via $W_s - \lambda_m \mathbf{I}_m = \tilde{W}$ to make it p.s.d. Since $\alpha \operatorname{tr}(U^T \mathbf{I}_m U) = \alpha d$ is constant for any scalar $\alpha$, the term $\lambda_m \mathbf{I}_m$ does not affect the optimisers of Problem (1). Thus, we can replace $W$ in (1) by the p.s.d. matrix $\tilde{W}$ without affecting the optimisers. $\qquad\square$

### Proof of Lemma 2

*Proof.* Consider the eigenvalue decomposition $(\Lambda, V)$ of $W$, i.e. $W = V\Lambda V^T$, where $\Lambda = \operatorname{diag}(\lambda_1, \ldots, \lambda_m)$ contains the decreasingly ordered (nonnegative) eigenvalues on its diagonal and $V \in \operatorname{St}(m,m)$. Consider the matrix $U \in \operatorname{St}(m,d)$, which can be written as the product of $V$ and another matrix $R \in \operatorname{St}(m,d)$, i.e., $U = VR$. So, instead of optimising over $U$, we can optimise over $R$. Let the $i$-th row of $R$ be denoted as $r_i$ for $i = 1, 2, \ldots, m$. It holds that $\operatorname{tr}(U^T W U) = \operatorname{tr}(R^T \Lambda R) = \sum_{i=1}^m \lambda_i \|r_i\|_2^2$. Thus, an equivalent formulation of Problem (1) is the optimisation problem

$$\max_{R=[r_1^T, r_2^T, \ldots, r_m^T]^T \in \operatorname{St}(m,d)} \quad \sum_{i=1}^m \lambda_i \|r_i\|_2^2. \tag{A10}$$

We observe that $0 \leq \|r_i\|_2^2 \leq 1$, and that $\sum_{i=1}^m \|r_i\|_2^2 = \operatorname{tr}(RR^T) = \operatorname{tr}(R^T R) = d$. Hence, a relaxation to (A10) is given by

$$\max_{p \in \mathbb{R}^m} \quad \sum_{i=1}^m \lambda_i p_i, \tag{A11}$$

$$\text{s.t.} \quad 0 \leq p_i \leq 1, \tag{A12}$$

$$\sum_{i=1}^m p_i = d. \tag{A13}$$

This is a linear programming problem for which an optimal solution is given by $p_1 = \ldots = p_d = 1$ and $p_{d+1} = \ldots = p_m = 0$ (since the $\lambda_i$'s are provided in decreasing order). Now we choose $R^* = [\mathbf{I}_d, \mathbf{0}]^T \in \mathbb{R}^{m \times d}$, so that

$$r_1^* = e_1^T, r_2^* = e_2^T, \ldots, r_d^* = e_d^T \quad \text{and} \quad r_{d+1}^* = \ldots = r_m^* = \mathbf{0}_d^T, \tag{A14}$$

where $e_i \in \mathbb{R}^d$ is the unit vector with element equal to 1 at the $i$-th place. We observe that for this choice $R^* \in \operatorname{St}(m,d)$, and that the objective value for (A10) is the same as the optimal value

for the problem defined by (A11)-(A13). Since the latter problem was a relaxation of the former problem, $R^*$ is an optimal solution to Problem (A10). The corresponding optimal $U$ for Problem (1) is $U^* = VR^* = V_d$. The observation that for any $U' = V_dQ$ with $Q \in \mathbb{O}(d)$ we have that $\text{tr}(U'^TWU') = \text{tr}(Q^TV_d^TWV_dQ) = \text{tr}(V_d^TWV_d)$ concludes the proof. $\square$

# B   Additional Experiments

In the following we provide further evaluations of our proposed algorithm.

## B.1   Step Size and Comparison to Two-Stage Approaches

In this section, on the one hand we experimentally confirm that our approach of choosing the step size $\alpha_t$ (see Sec. 4.3) is valid and that in practice it is not necessary to perform line search. On the other hand, we verify that our proposed algorithm leads to results that are comparable to two-stage approaches derived from Lemma 4. Such two-stage approaches first determine the matrix $U_0 \in \text{im}(V_d)$ that spans the $d$-dimensional dominant subspace of $W$, and subsequently utilise the updates in equations (2) and (3) in order to make $U_0$ sparser. As explained, this corresponds to finding a matrix $Q \in \mathbb{O}(d)$ that maximises our secondary objective

$$g(U_0Q) = \sum_{i=1}^{m}\sum_{j=1}^{d}(U_0Q)_{ij}^p. \tag{A15}$$

We compare our proposed algorithm to two different settings of two-stage approaches:

1. **Our algorithm as stage two.** Our proposed algorithm forms the second stage of a two-stage approach. To this end, in the first stage we use the Orthogonal Iteration algorithm [21] to find the matrix $V_d$ that spans the $d$-dimensional dominant subspace of $W$. Subsequently in the second stage, we initialise $U_0 \leftarrow V_d$, and according to Lemma 4 we make use of the updates in equations (2) and (3) in order to make $U_t$ iteratively sparser. We consider two variants for the second stage:

   (a) In the variant denoted OURS/2-STAGE we run the second-stage updates exactly for the number of iterations that the Orthogonal Iteration required in the first stage to find $V_d$ (with convergence threshold $\epsilon = 10^{-5}$). We use the step size $\alpha_t$ as described in Sec. 4.3.

   (b) In the variant denoted OURS/2-STAGE/BT we utilise backtracking line search (as implemented in the ManOpt toolbox [14]) in order to find a suitable step size $\alpha_t$ in each iteration. Here, we run the algorithm until convergence w.r.t. to $g$, i.e. until $g(U_tQ_t)/g(U_{t+1}Q_{t+1}) \geq 1 - \epsilon$ for $\epsilon = 10^{-5}$.

2. **Manifold optimisation as stage two.** Further, we consider the trust regions method [2] to find a (local) maximiser of (A15) in the second stage. Here, the optimisation over the Riemannian manifold $\mathbb{O}(d)$ is performed using the ManOpt toolbox [14]. For the first stage, we consider three different initialisations for finding the matrix $V_d$ that spans the $d$-dimensional dominant subspace of $W$: the Matlab functions *eig()* and *eigs()*, as well as our implementation of the Orthogonal Iteration algorithm [21]. We call these methods EIG+MANOPT, EIGS+MANOPT and ORTHIT+MANOPT, respectively.

Results are shown in Figs. 4 and 5 for the CMU house sequence and the synthetic dataset, respectively. We observe the following:

- In terms of **solution quality** (fscore and objective), all considered methods are comparable in most cases. For the real dataset (Fig. 4) EIGS+MANOPT performs worse due to numerical reasons. For the largest considered permutation synchronisation problems (the right-most column in the synthetic data setting shown in Fig. 5) OURS leads to the best results on average.

- In terms of **runtime**, in overall OURS is among the fastest, considering both the real and the synthetic data experiments. In the real dataset, where $d = 30$ is relatively small, EIGS+MANOPT is the fastest (but with poor solution quality), while EIG+MANOPT is the

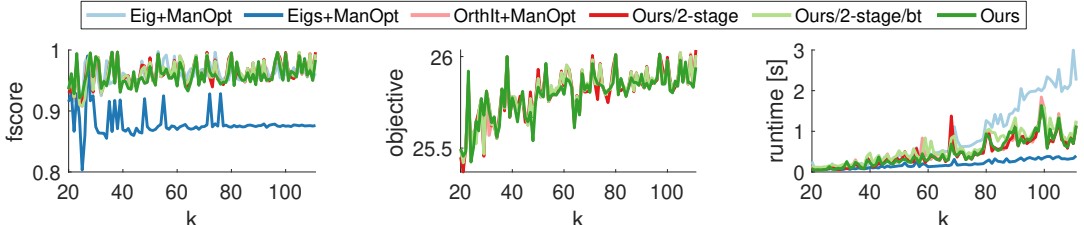

Figure 4: Comparison of OURS to different two-stage approaches on permutation synchronisation problems from the CMU house sequence (see Sec. 4.3 for details). We consider the fscore (↑), objective value (↑), and runtime (↓). The individual instances of permutation synchronisation problems vary along the horizontal axis.

slowest (with comparable solution quality to OURS). Methods that utilise the Orthogonal Iteration have comparable runtimes in the real data experiments.

Most notably, in the largest considered synthetic data setting (right-most column in Fig. 5) OURS is among the fastest (together with OURS/2-STAGE), while OURS has the largest fscore on average (as mentioned above) – this indicates that OURS is particularly well-suited for permutation synchronisation problems with increasing size.

- Overall OURS is the simplest method, see Algorithm 1: the solution is computed in one single stage rather than in two consecutive stages, and it does not require **line search**, as can be seen by comparing OURS with OURS/2-STAGE/BT across all experiments.

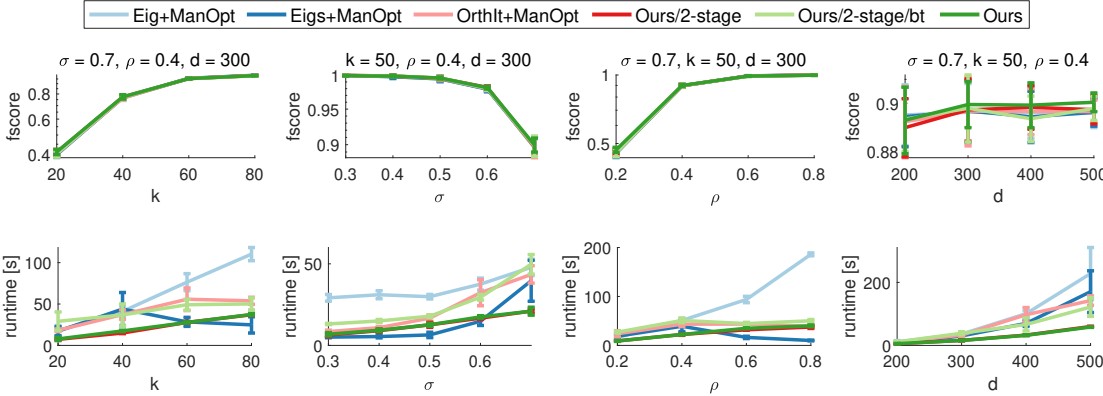

Figure 5: Comparison of OURS to different two-stage approaches on synthetic permutation synchronisation problems (see Sec. 4.3 for details). Each column shows a different varying parameter. The first row shows the fscore (↑) and the second row the runtime (↓). Note that the right-most column shows the largest considered permutation synchronisation instances – for these OURS obtains the best fscore while being among the fastest (together with OURS/2-STAGE).

### B.2 Comparison to Riemannian Subgradient and Evaluation of Different $p$

In Fig. 6 we compare OURS with $p = 3$ and $p = 4$ to the Riemannian subgradient-type method (with QR-retraction) by Li et al. [27] with $\ell_1$-norm as sparsity-inducing penalty. In the qualitative results (bottom) we can observe that the Riemannian subgradient-type method and Ours ($p = 4$) obtain sparse solutions with few elements with large absolute values (both positive and negative), whereas Ours ($p = 3$) obtains a sparse and (mostly) nonnegative solution. Since for permutation synchronisation we are interested in nonnegative solutions, OURS with $p = 3$ thus outperforms the two alternatives quantitatively (top).

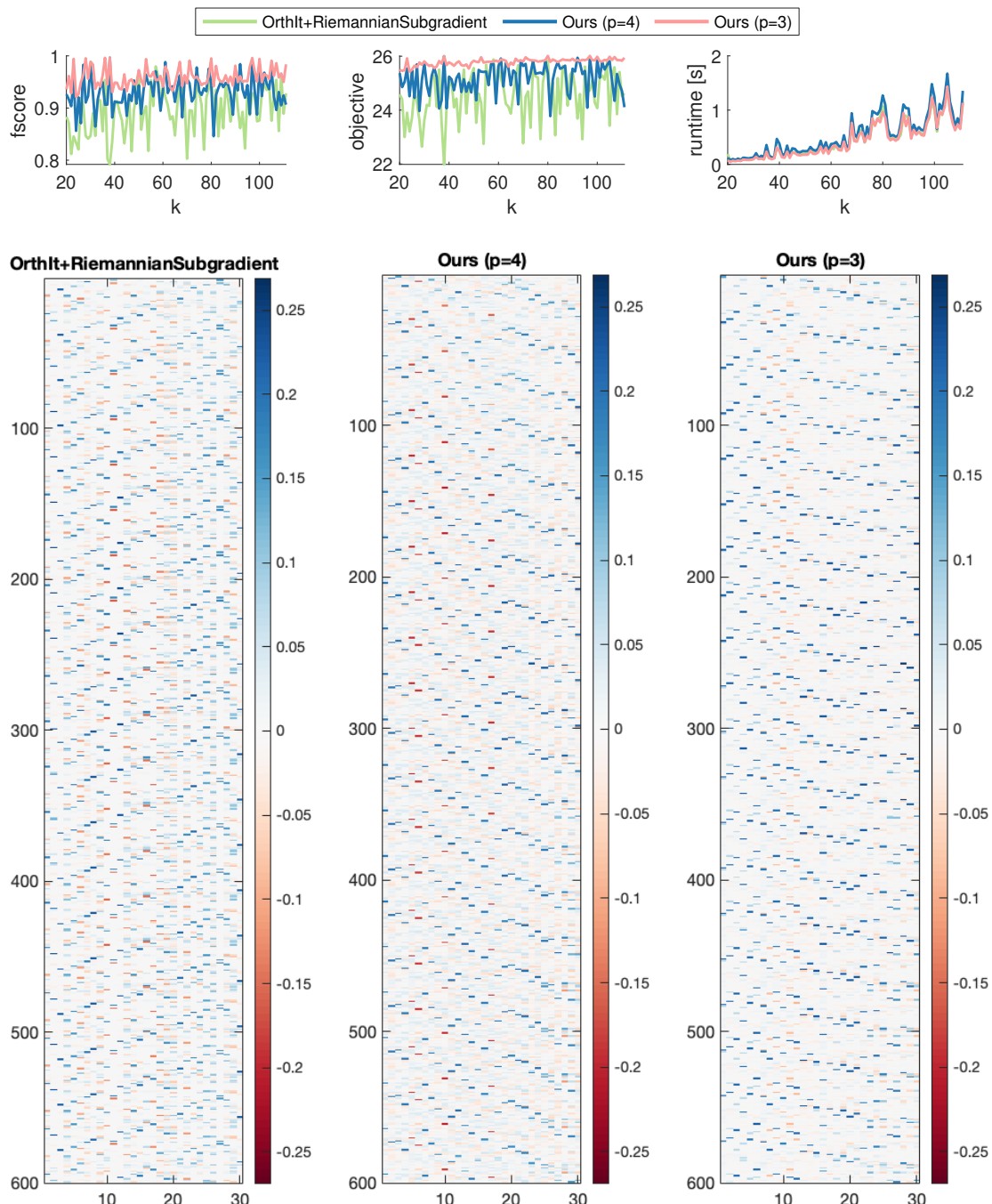

Figure 6: Comparison of OURS (with $p = 3$ and $p = 4$) to the Riemannian subgradient-type method by Li et al. [27]. Here, permutation synchronisation problems from the CMU house sequence (see Sec. 4.3 for details) are evaluated. **Top:** we consider the fscore ($\uparrow$), objective value ($\uparrow$), and runtime ($\downarrow$), where the individual instances of permutation synchronisation problems vary along the horizontal axis. **Bottom:** for each of the three methods we show the obtained $U$-matrix for $k = 20$ (before projection). It can be seen that the Riemannian subgradient-type method and Ours ($p = 4$) obtain sparse solutions with few elements with large absolute values (both positive and negative), whereas Ours ($p = 3$) obtains a sparse and (mostly) nonnegative solution.

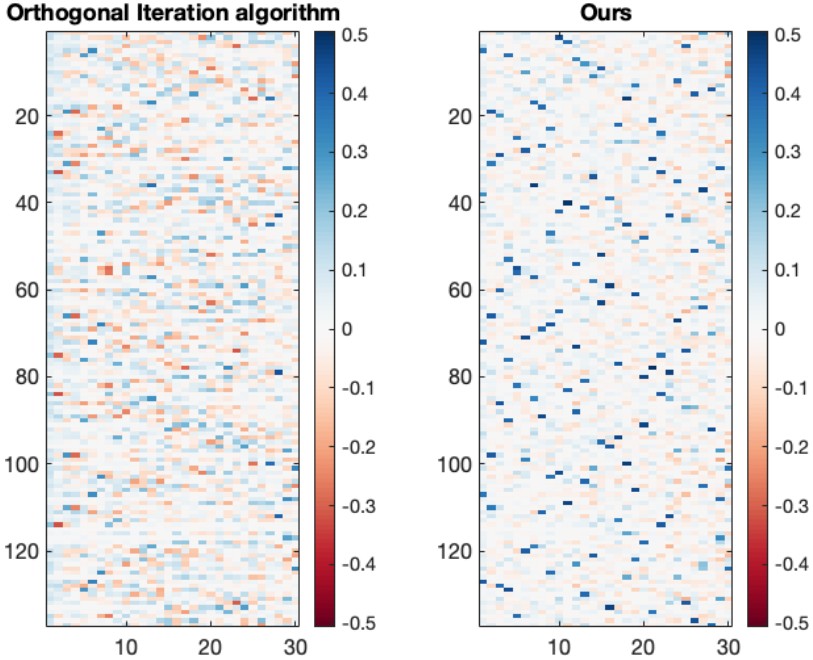

Figure 7: Illustration of the effect of our sparsity-promoting secondary objective $g$ for a synthetic permutation synchronisation problem ($k = 5, d = 30, \rho = 0.9, \sigma = 0.3$, cf. Sec. 4.3). The matrix $U$ obtained by the Orthogonal Iteration algorithm (left) is not sparse. Our method gives a sparse and mostly nonnegative $U$ (right).

## B.3 Effect of Sparsity-Promoting Secondary Objective

In Fig. 7 we illustrate the effect of our sparsity-promoting secondary objective. It can clearly be seen that our method (right) results in a significantly sparser solution compared to the Orthogonal Iteration algorithm (left).