# OpenReview forum: "Sparse Quadratic Optimisation over the Stiefel Manifold with Application to Permutation Synchronisation"
_NeurIPS.cc/2021/Conference — NeurIPS 2021 Poster_

### Official Review · Reviewer_C7cB · 2021-07-14

**Rating:** 4
**Confidence:** 5

**Summary:**

The paper is concerned with pursuing sparse optimal solutions to optimization problems of the form max Trace(U^T W U) [*] where U is an orthonormal matrix, that is U lies in the Stiefel manifold. This problem is meaningful since it typically arises as a relaxation over the Stiefel manifold of combinatorial problems such as spectral clustering and permutation synchronization. It this latter problem that occupies the application aspect of the paper. A global (not necessarily sparse) solution to [*] can be obtained in closed form from the eigendecomposition of W. Algorithmically, this can be computed by the power iteration algorithm. The paper proposes a variation of the power iteration algorithm (Algorithm 1), which includes a sparsity promoting step, forcing the basis matrix U to be close to a power of U. Permutation synchronization experiments with synthetic data and on the CMU "house" sequence for finding correspondences in two views,  show that this approach gives better permutation synchronization than several other existing alternatives.

**Ethical Concerns:**

No issues as far as I can see.

**Limitations And Societal Impact:**

No issues as far as I can see.

**Main Review:**

The idea of forcing sparsity on U towards recovering a solution of the original problem (the one whose relaxation is [*]) is interesting and important. Moreover, the idea of modifying the power iteration method with the sparsity promoting step of (4) is interesting as well.

On the other hand, a weakness in the paper is that the authors do not treat the attribute "sparse" precisely.  In page 4, line 136, when they introduce their algorithm, they say that they are interested in an optimal solution of [*] which is sparse "in some relaxed sense". Only in the discussion section at page 7 they write "In turn, we rather interpret sparsity in some looser sense, meaning that there are few elements that are large, whereas most elements are close to zero." I feel that this should have been clarified very early on.

Another weakness is that the authors do not provide any experimental analysis of Algorithm 1 alone.
I run the code of the authors on a very simple case, where W=diag(1,1,1,0.5,0.5,0.5), with d=3 and various values of p ranging from 2 to 20. Ideally, the algorithm should be converging to the matrix E=[e1, e2, e3] where e1,e2,e3 are the first three standard basis vectors of R^6. The algorithm converges very quickly (in two iterations or so) to a matrix that is far from E (in fact most elements in the matrix seem to have the same order of magnitude), but if one only selects the largest in magnitude element per column and sets the rest to zero, then one does get E M, where M is a diagonal matrix (this is more clear for larger values of p). More precisely, if one runs the linear assignment post-processing step that the authors propose, then we get back E for p=10, but a wrong result for p=2,3. This shows that both 1) the linear assignment step  and 2) the value of p, are crucial. I also did another experiment at the other extreme, where I chose W to be a generic 6x6 positive semi-definite matrix. With d=5 one expects 4+3+2+1 zeros (this is the highest sparsity of an orthonormal basis of a generic 5-dimensional subspace of R^6). With p=10, while again most elements have the same order of magnitude, there is indeed a clear cluster of 10 smaller elements.

In conclusion, Algorithm 1 is an interesting heuristic, but more investigation is needed.
Since it is so simple, perhaps it is possible to actually provide a precise mathematical analysis. For this, the authors should define their approximate notion of sparsity rigorously and prove that if the d-th principal eigenspace of W admits an orthonormal basis of sparsity level \alpha, then for p large enough, Algorithm 1 converges to an orthonormal basis with approximate sparsity \alpha.
I believe this would make a strong paper.

Three more suggestions:
1) I think it would be more interesting to apply permutation synchronization to a less explored and modern application, such as record linkage, instead of feature matching in computer vision, where there is a vast literature.
2) Lemmas 1 and 2 are textbook level, there is no need to include the proofs (in fact the given proof of Lemma 2 is not complete).
3) The notation U* \in im(bar(U)) is not precise.

**** Post-Rebuttal ****

I thank the authors for their responses. I would like to keep my rating as is, in anticipation of a stronger version of this work.

**Time Spent Reviewing:**

14

---

> ### Author Response · Authors · 2021-08-06
> **The mentioned issue was due to a too early termination of our algorithm**
>
> We are happy that the reviewer finds our “idea [...] interesting and important”. We want to thank the reviewer for the detailed and constructive feedback, and for bringing up several suggestions to further improve our work. In the following, we address the reviewer’s main concerns:
>
> **1. Notion of “sparsity”**
>
> At the beginning of Sec. 3 we will make it clear that by “sparsity” we mean “that there are few elements that are large, whereas most elements are close to zero”, and that we will consider Eq. (4) to quantify sparsity.
>
> **2. Experimental analysis of Alg. 1**
>
> Thank you for taking the time to experiment with our algorithm. We believe that the mentioned issue is neither due to the specific choice of $p$, nor that our method strongly depends on the post-processing. Instead, the encountered issue stems from terminating too early. This happens since the current termination criterion in our algorithm only considers the primary objective $f(U)$, and for this simple example the sequence $\\{f(U_t)\\}$ converges already after two iterations. However, two iterations are insufficient for our secondary objective $g(U)$ to appropriately sparsify the solution.
>
> For the permutation synchronisation problem, which is the main motivating application of this work, we experimentally confirmed that using the primary objective as sole termination criterion is sufficient in practice. This can be seen in Figs. 1 and 2 in the supplementary document when comparing “Ours” vs. “Ours/2-Stage/BT”. For the latter, we also consider the secondary sparsifying objective as termination criterion (see l23-24 in the supplementary document). In this case, using the step size $\alpha_t = || h - h^T ||^{-1}$ (as explained in l252 main paper) is not possible anymore, since this choice does not guarantee a monotonic increase of the secondary objective (so that using it as termination criterion does not make sense). Hence, for the setting “Ours/2-Stage/BT” backtracking line search is used.
>
> To confirm these elaborations, here https://www.dropbox.com/s/n5f72zbg72ielu1/SparseStiefelOpt2.zip?dl=0 (anonymous link for double-blind review) we provide updated code that reproduces the issues faced by the reviewer. Additionally, the code implements a termination criterion that considers both the primary and secondary objective at the same time. In this case, various choices of $p \geq 3$ lead to a sparse solution that solve the mentioned issue (note that for even $p$ the columns of $U$ can also be negative; moreover, since we do not guarantee global optimality in the secondary objective, there is a possibility that the secondary objective does not converge to a global optimum).
>
> We will reflect these points in our manuscript as follows:
>
> a) In Alg. 1 we will change the convergence criterion to a general “until convergence”
>
> b) In Sec. 4.3 we will explain that we use the relative improvement of the primary objective $f(U_t)/f(U_{t+1}) \geq 1- \epsilon$ as convergence criterion for all permutation synchronisation experiments (hence, all experiments remain as they are).
>
> c) We will provide implementations of both convergence criteria in the code we release, and we will also include the simple example in our code.
>
> d) In Sec. 5 we will complement the statement “Studying the universality of the proposed method and [...] are open problems that we leave for future work.” with an explanation regarding the convergence criteria, as explained above.
>
> **3. Mathematical analysis of Alg. 1**
>
> We absolutely agree that guarantees or bounds with respect to the achieved sparsity would be a strong addition to our manuscript. However, due to the non-convexity of our secondary problem we cannot in general expect to find global optima. As such, even if $W$ admits a certain level of sparsity, we cannot guarantee that this level of sparsity will be achieved from an arbitrary initialisation.
>
> **4. Application to record linkage**
>
> Thank you for bringing the record linkage problem as an application of permutation synchronisation to our attention. We will consider this as an alternative application in potential follow-up works.
>
> **5. Lemmas 1 & 2**
>
> We included Lemma 1 and 2 for the sake of completeness. We will add the missing parts to the proof of Lemma 2, and then move both proofs to the supplementary material (as suggested by reviewer 9Pkc).
>
> **6. Notation**
>
> We will make the notation more rigorous by writing $U^* \in \\{ U \in \text{St}(m,d) : \text{im}(U) = \text{im}(\bar{U}) \\}$.

---

### Official Review · Reviewer_9Pkc · 2021-07-16

**Rating:** 6
**Confidence:** 5

**Summary:**

This paper proposes a novel algorithm to solve the permutation synchronization problem which can be formulated as minimizing a quadratic function over the Stiefel manifold. The algorithm is modified from the Orthogonal Iteration method with sparsity promotion. The proposed algorithm is shown to perform better than existing methods in terms of both efficiency and accuracy. Moreover, the limitation  of the work  is also discussed.

**Limitations And Societal Impact:**

Limitations:
The limitations have been discussed by the authors.

Societal Impact:
Not apply.

**Main Review:**

1. The proposed algorithm has strong intuitions. And the presentation of this work is easy to follow.   To promote the sparsity, the objective function $g(UQ)$ is introduced. The authors point out that $g(UQ)$ is a relaxation of the sparsity and non-negativity.    Since $g(UQ)$ is nonconvex, the step (7) may converge to the local minimum points of (5). The related discussions are given in Appendix. It's better to mention this in the main paper.

2. There is also a post-processiong step after algorithm 1  as stated in Section 4.2. It would be better to put them together in the Algorithm environment.

3. The reviewer is curious about the numerical results of the case $p=4$ and minimizing the following function $\sum_{ij} |(UQ)_{ij}|$. For the latter nonsmooth function, you may try  the Riemannian subgradient method [Li. et al 2021] with QR retraction.

Minor:
1. The proofs of Lemma 1 and 2 are trivial. I suggest putting them in the Appendix.
2. Line 138: 'solution to (1)'

Reference:
[Li. et al 2021] Xiao Li, Shixiang Chen, Zengde Deng, Qing Qu, Zhihui Zhu, Anthony Man-Cho So. Weakly Convex Optimization over Stiefel Manifold Using Riemannian Subgradient-Type Methods. SIAM Journal on Optimization (2021) 31(3):1605-1634.

-----------------------------------------------------------------------
Post rebuttal:
The authors sucessfully address my concerns.

**Time Spent Reviewing:**

4

---

> ### Author Response · Authors · 2021-08-06
> **Comparison to Riemannian Subgradient**
>
> We thank the reviewer for taking the time to carefully read our paper and for providing constructive feedback that will help to further improve our work. We are happy to read that the reviewer agrees that our “algorithm has strong intuitions”, and that the “work is easy to follow”.
>
> **1.	Non-convexity of secondary objective**
>
> Thank you for the suggestion. We will include a discussion regarding the non-convexity of the secondary objective $g$ in the main paper.
>
> **2. Postprocessing in Alg. 1**
>
> Since the projection is not part of our theoretical analysis, we find it is more appropriate to not include it explicitly into Alg. 1. If requested, we are happy to include an informal statement about it in the caption of Alg. 1.
>
> **3. Numerical results**
>
> We thank the reviewer for the suggestion of evaluating our method with $p=4$ and the Riemannian subgradient method by Li et al. with the $\ell_1$-norm as penalty for rotating $U$ via the orthogonal matrix $Q$. We have conducted an additional experiment for various permutation synchronisation problem instances. Since both, our method with $p=4$, as well as the $\ell_1$-norm penalty, induce sparsity, but do not favour nonnegative solutions, after optimisation with either method we flip the signs of the columns of the obtained $U$ in such a way that the column sums are nonnegative. Results can be found here: https://www.dropbox.com/s/x1izvb4ximwf9u3/evaluation_riemannian_subgradient.png (anonymous link for double-blind review). It can be clearly seen that our method with $p=3$ outperforms the two others in terms of the fscore and the objective value, while all methods have a comparable runtime. We will include these results in the appendix.
>
> **4. Minor points**
>
> We will move the proofs of Lemmas 1 and 2 to the appendix, and update l138.

---

### Official Review · Reviewer_ZQ5Z · 2021-07-17

**Rating:** 6
**Confidence:** 3

**Summary:**

In this paper the authors propose a modification of a well-known algorithm, in order to maximize a quadratic function over the Stiefel manifold, promoting sparsity at the same time. The base method is not a gradient-retraction over a manifold algorithm (although some tools are used in the middle), but a QR type of method, specifically the Orthogonal Iteration algorithm. The modification is the addition of a right multiplicative matrix, which is selected in order to promote sparsity. This is done for a maximization problem, and the $l_p$ norm (with $p>2$) is used to that end. The method is tested for the permutation synchronization problem.



**Limitations And Societal Impact:**

The limitations are correctly stated in Section 5, but specially the one regarding the equal eigenvalues is easy to overlook in the rest of the text.

**Main Review:**

The method is a clever modification of a very well-known algorithm. The novelty is the product with a matrix $Z(U_t)$, which doesn't affect the convergence of the original method, and at the same time it allows to incorporate a second objective function, which is chosen as the $l_p^p$ norm of the vectorized matrix.
The existing literature is correctly described, and the difference with respect to existing approaches is discussed.


The presented theoretical results seem to be correct. The proofs of the first part are simple, using basically linear algebra tools, but this is sufficient to prove the convergence and the bahaviour of the method in general.

The paper is correctly written, and it provides sufficient elements to reproduce the results. I wouldn't say is an "easy to follow" paper. I had to go back and forth several times in order to understand some parts. At one point I kept reading even though I was not completely understanding the goal of some points, with the hope to understand it later, which I think I did. I think this is in part due to the method itself, and not fault of the authors.

I think that the contributions are significant, in the sense that the problem is relevant, this method seems to perform well for some difficult tasks, and the modifications are simple but clever. I'm not sure if those ideas could be used for other methods, not based on the QR methodology. Maybe they can.

Some comments and minor comments:
In the experimental section, I would have liked to see an experiment showing the effect of the sparsity promoting addition. Maybe comparing the sparsity of resulting matrix from the original Orthogonal Iteration algorithm (which will be dense most likely), with sparsity resulting from your method.

In lines 112-113, the sentence might be read as if $l_p$ norms with $p<1$ are smooth (I know the observation is about the convexity).

Post rebuttal: The authors addressed my concerns


**Time Spent Reviewing:**

3

---

> ### Author Response · Authors · 2021-08-06
> **Effect of sparsity-promoting term**
>
> We thank the reviewer for taking the time to carefully read our paper, and for providing constructive feedback that will further improve our manuscript. We are happy that the reviewer acknowledges that our “method is a clever modification of a very well-known algorithm”, and that our “contributions are significant”. In the following we comment on the requests/concerns that the reviewer brings up:
>
> **1. Effect of sparsity-promoting term**
>
> The effect of the sparsity-promoting term for a synthetic permutation synchronisation instance can be found here: https://www.dropbox.com/s/mb7bw1s3cjxyhcs/ablation_sparsity.png (anonymous link for double-blind review). It can be clearly seen that our method results in a significantly sparser solution. We will include this illustration in the appendix.
>
> **2. Ambiguous formulation**
>
> To avoid ambiguities, we will clarify the sentence in l112-113 as follows: “In order to promote sparse solutions, sparsity-inducing regularisers can be utilised, for example via the minimisation of the (non-convex) $\ell_p$-'norm' for $0 \leq p < 1$, or the (convex) $\ell_1$-norm [23]. However, the non-smoothness of such regularisers often constitutes a computational obstacle.”
>
> **3. Equal eigenvalue assumption**
>
> The equal eigenvalue assumption is currently stated in Lemma 4, the sentence before Lemma 4, and in Sec. 5. Yet, to avoid that it is overlooked, we will in addition remind the reader about it in Sec. 3.2 (l199).

---

### Official Review · Reviewer_fYRj · 2021-07-17

**Rating:** 7
**Confidence:** 5

**Summary:**

They address sparse quadratic optimization over the Stiefel manifold with application to permutation synchronization.

**Limitations And Societal Impact:**

Yes.

**Main Review:**

Existing solvers, e.g. based on eigenvalue decomposition, are unable to account for sparsity while at the same time maintaining global optimality guarantees. They fill this gap and propose a simple yet effective sparsity-promoting modification of the Orthogonal Iteration algorithm for finding the dominant eigenspace of a matrix.

**Time Spent Reviewing:**

10

---

> ### Author Response · Authors · 2021-08-06
> **Thank you**
>
> We are happy that the reviewer considers our work a “good paper”. Since no further clarifications are requested, we would like to use this opportunity to thank the reviewer for taking the time to carefully review our manuscript.

---

### Decision · Program_Chairs · 2021-09-27

**Decision:**

Accept (Poster)

**Comment:**

The proposed method for sparse quadratic optimization over the Stiefel manifold and its analysis are interesting. However, the technical presentation and theoretical results have to be strengthened.